# Garlic Volatile Diallyl Disulfide Induced Cucumber Resistance to Downy Mildew

**DOI:** 10.3390/ijms222212328

**Published:** 2021-11-15

**Authors:** Fan Yang, Hui Wang, Chengchen Zhi, Birong Chen, Yujie Zheng, Lijun Qiao, Jingcao Gao, Yupeng Pan, Zhihui Cheng

**Affiliations:** College of Horticulture, Northwest A&F University, Xianyang 712100, China; yangfan3@nwafu.edu.cn (F.Y.); wang_hui@nwafu.edu.cn (H.W.); zhichengchen@nwafu.edu.cn (C.Z.); chenrrr@nwafu.edu.cn (B.C.); zyujie@nwsuaf.edu.cn (Y.Z.); qiaolijun@nwafu.edu.cn (L.Q.); gaojingcao@nwsuaf.edu.cn (J.G.)

**Keywords:** cucumber, downy mildew, diallyl disulfide, H_2_O_2_, plant hormone, transcriptome

## Abstract

Allicin compositions in garlic are used widely as fungicides in modern agriculture, in which diallyl disulfide (DADS) is a major compound. Downy mildew, caused by *Pseudoperonospora cubensis* (*P. cubensis*), is one of the most destructive diseases and causes severe yield losses in cucumbers. To explore the potential mechanism of DADS-induced cucumber resistance to downy mildew, cucumber seedlings were treated with DADS and then inoculated with *P. cubensis* at a 10-day interval. Symptom observation showed that DADS significantly induced cucumber resistance to downy mildew. Furthermore, both lignin and H_2_O_2_ were significantly increased by DADS treatment to responding *P. cubensis* infection. Simultaneously, the enzyme activities of peroxidase (POD) in DADS-treated seedlings were significantly promoted. Meanwhile, both the auxin (IAA) and salicylic acid (SA) contents were increased, and their related differentially expressed genes (DEGs) were up-regulated when treated with DADS. Transcriptome profiling showed that many DEGs were involved in the biological processes of defense responses, in which DEGs on the pathways of ‘phenylpropanoid biosynthesis’, ‘phenylalanine metabolism’, ‘MAPK signaling’, and ‘plant hormone signal transduction’ were significantly up-regulated in DADS-treated cucumbers uninoculated with the pathogen. Based on the results of several physiological indices and transcriptomes, a potential molecular mechanism of DADS-induced cucumber resistance to downy mildew was proposed and discussed. The results of this study might give new insight into the exploration of the induced resistance mechanism of cucumber to downy mildew and provide useful information for the subsequent mining of resistance genes in cucumber.

## 1. Introduction

Garlic (*Allium sativum* L.), an important economic vegetable, contains various sulphureous volatile organic compounds (VOCs), including sulfides, disulfides, and trisulfides, all of which are formed by the decomposition of allicin and are released upon crushing garlic [1]. These sulphureous VOCs give garlic strong allelopathic effects, which makes garlic a popular targeted crop in rotation and intercropping systems. Until now, garlic has been rotated or intercropped with many other vegetable crops such as eggplant [2], pepper [3], tomato [4], and cucumber [5,6]. When garlic is used in the intercropping systems, several beneficial effects have been reported [7], such as improving the nutritional status of soil, reducing plant diseases and pests, alleviating continuous cropping obstacles, and increasing land use efficiency and net return. Wang et al. [2] reported that eggplant intercropped with garlic could be an ideal farming system to effectively improve soil nutrients and to partially alleviate soil-borne diseases. In the pepper–garlic intercropping system, garlic root exudates with lower concentrations can stimulate the protective enzyme systems and then promote the growth of pepper, while those with higher concentrations will show deleterious effects [3]. For the tomato–garlic intercropping system, the increased microorganism populations and enzyme activities in the substrates were observed, which further improved the fruit quality of tomato and produced higher economic benefits than the mono-cultivated tomatoes [4]. In addition, compared with the monoculture system, both the cucumber–garlic and cucumber–green garlic intercropping systems showed improving effects on the populations of microorganisms and the activities of soil enzymes, as well as the nutritional status of both soils and plants [5,6]. Our previous studies indicated that intercropping green garlic or garlic bulbs with cucumbers can significantly reduce the incidence and disease index of downy mildew in cucumber at the early stage of onset, and intercropping with green garlic had more significant effects. In the cultivation system of green garlic intercropped with cucumber, the VOCs released from green garlic might play important roles in reducing the incidence of downy mildew in cucumber; especially for the period of harvesting green garlics, more intense VOCs will be released. However, the regulating mechanism of these green garlic released VOCs in inducing cucumber downy mildew resistance is still unknown.

Diallyl disulfide (DADS) is a major allelochemical of the VOCs in garlic [1,8]. Previous studies found that DADS can promote root growth in tomato by affecting cell division, changing endogenous phytohormones, increasing expressions of expansion genes [8]. Additionally, the pathways of sulphate assimilation and glutathione (GSH) metabolism are also involved in the regulation of DADS-induced tomato root growth [9]. Recently, Cheng et al. reported that DADS might increase resistance and promote plant growth in tomato by increasing its root activity and photosynthetic capacity and development to reduce the autotoxicity of tomato [10]. In addition, lower concentrations of DADS can promote root growth and induce the elongation of main roots in cucumber [11]. Our previous work indicated that both the green garlic VOCs and DADS play vital roles in the reactive oxygen species (ROS) production and regulation of antioxidant enzymes in cucumber seedlings [1]. Moreover, DADS was also reported to have antifungal activity against several fungi (e.g., *C. albicans*, *C. tropicalis*, and *Blastoschizomyces capitatus*) [12]. Therefore, the DADS might also play an important role in the inducing of disease resistance in cucumber.

Downy mildew, caused by the obligate oomycete pathogen *P. cubensis*, is one of the most devastating diseases in cucumber [13]. Although some resistant germplasms have been identified, the regulating genes and the pathways involved in mediating downy mildew resistance in cucumber are still unclear [14]. Several quantitative trait loci associated with downy mildew resistance have been mapped onto five of the seven cucumber chromosomes, suggesting that downy mildew resistance in cucumber is likely due to the associative effects of multiple genes [15]. Recently, *CsSGR* was reported as a major-effect resistant gene for cucumber downy mildew. Loss-of-function of *CsSGR* gives cucumber durable and broad-spectrum disease resistance to several diseases including the downy mildew. This probably is due to the fact that *CsSGR* functions in the inhibitions of ROS over-accumulation and phytotoxic catabolite over-buildup in the chlorophyll degradation pathway [16]. Additionally, overexpression of *CsWRKY50* also enhanced cucumber resistance to *P. cubensis*, probably, through upregulating the transcript levels of several phytohormone-related defense genes that include salicylic acid (SA)-responsive genes, jasmonic acid (JA) responsive genes, and SA biosynthesis genes [17]. Furthermore, transcriptome profiling in the context of *P. cubensis* infection also gives some insight into understanding the molecular mechanisms of host responses to downy mildew infection. In cucumber, transcriptome data revealed that resistance to downy mildew is associated with an earlier response to the pathogen, hormone signaling, and regulation of nutrient supply [14]. Additionally, the pathways of pathogen-associated molecular pattern (PAMP) recognition, signal transduction, ROS and lignin accumulation, and some transcription regulators are all involved in the resistance of cucumber to downy mildew [15]. Although some research on germplasm identification, regulatory genes, and pathways of cucumber resistance to downy mildew were performed, studies on the use of allelopathic substances to induce cucumber resistance to downy mildew are rarely reported, and its potential regulatory mechanism is also unknown.

Therefore, based on the previous studies, we speculate that DADS may play an important role in promoting the cucumber resistance to downy mildew. To explore the potential mechanism of DADS-induced cucumber resistance to downy mildew, phenotypic observation, physiological changes, and transcriptome analysis were performed to reveal the infection processes and cucumber resistance responding mechanisms to the infection of *P. cubensis*. Results of this study lay a foundation for research on the resistance mechanism of downy mildew and provide useful information for the subsequent mining of resistance genes in cucumber. Additionally, this work might provide an alternative environmentally friendly way for reducing the continuous cropping obstacles and controlling leaf diseases during the cultivation of vegetables.

## 2. Results

### 2.1. DADS-Induced Cucumber Resistance to Downy Mildew

To investigate the effect of DADS on *P. cubensis* infection, the phenotypic changes of cucumber seedlings under the inoculation of pathogens were observed and recorded. For the seedling of CK, the pathogen-inoculated area firstly showed a water-soaking phenotype and then developed chlorotic lesions (Figure 1A and Appendix A). However, the DADS-treated seedlings showed smaller, less chlorotic lesions (Figure 1A and Appendix A). Statistically, the disease index of DADS treatment (DI = 17.2) was significantly lower than that of CK (DI = 54.3) (*p* < 0.01) (Figure 1B). Additionally, although the leaves of DADS-treated seedlings developed typical disease spots and formed apparent necrotic areas, these symptoms did not continuously become larger or worse after 14 days post-inoculation (Appendix A). In contrast, more disease spots and necrotic areas were observed on the leaves of CK seedlings, in which these symptoms continued to grow larger even after 14 days inoculation (Appendix A). Briefly, compared with the CK seedlings, the DADS-treated cucumber seedlings showed better resistance to downy mildew based on the occurred symptoms.

Microscopic examination of the growth process of *P. cubensis* among the infected lesions also showed consistent results with the observed phenotypic changes. The sporangiophore occurred as early as 24 hpi (hours of post inoculation) in CK seedlings, while the DADS-treated seedlings had not shown sporangiophore at this time point (Figure 1C). At 96 hpi, many sporangiophores grew in CK seedlings, but it was not observed in DADS-treated seedlings. Sporangiophore development occurred with a lot of sporangium in CK seedlings at 168 hpi, but these phenomena had not occurred in DADS-treated seedlings. In general, the growth of *P. cubensis* in the DADS treated seedling were relatively slower and abnormal (less sporangiophore and rare sporangium) compared with that in the seedlings of CK, which might further result in the DADS-treated cucumber seedlings with resistance features to downy mildew.

### 2.2. DADS Induced More ROS and Lignin Accumulations and Defense Enzymes Divergences after P. cubensis Inoculation

The production of cellular ROS, including H_2_O_2_ and superoxide anion, can be induced by pathogen invasion [18]. To verify whether ROS accumulation was associated with the DADS induced cucumber resistance to *P. cubensis*, the contents of H_2_O_2_ and superoxide anion in the leaves of DADS-treated seedlings were checked. DADS-treated leaves showed a similar changing trend with CK for the content of superoxide anion, but the contents in DADS-treated leaves were higher than that in CK, especially for the time points of 24, 48, and 72 hpi (Figure 2A). Either inoculated or not, H_2_O_2_ was found in both DADS treatment and CK, as manifested by quantitative measurement and histochemical staining with DAB (Figure 2B,C). DADS-treated leaves accumulated significantly higher levels of H_2_O_2_ at 0, 4, and 12 hpi than those of CK leaves (Figure 2B,C). However, DADS-treated leaves accumulated significantly lower levels of H_2_O_2_ than those of CK leaves after 24 hpi (Figure 2C). In DADS-treated leaves, a dramatic increase in H_2_O_2_ content was observed at 4 hpi, while a second slight increase was detected between 24 hpi and 48 hpi (Figure 2C). In CK leaves, only a single increase was observed, peaking at 72 hpi (Figure 2C). At the final monitored time point of 168 hpi, H_2_O_2_ content in the DADS treated leaves was lower than that in CK (Figure 2C).

Since the biosynthesis of H_2_O_2_ was mainly controlled by SOD, and its degradation was regulated by POD and CAT, enzymatic assays were performed to firstly check the activities of these three enzymes (SOD, CAT, and POD). Additionally, the activities of PAL, the first enzyme in the phenylpropanoid metabolism pathway [19], were also examined. The results showed that the activity of SOD in DADS-treated leaves was significantly higher than that in CK leaves at 24 hpi, while significantly lower activities were shown in DADS treated leaves at both 48 and 168 hpi (Figure 2D). The activities of CAT in DADS-treated leaves were significantly higher and lower than those in CK leaves at both 48 and 168 hpi and both 12 and 24 hpi, respectively (Figure 2E). DADS-treated leaves showed continuously higher POD activities than those in CK leaves at 4, 12, 48, and 72 hpi (Figure 2F). The activities of PAL in DADS-treated leaves were lower than those in CK leaves, especially for the time points of 48, 72, 96, and 168 hpi (Figure 2G). Collectively, considering the enzyme corresponding functions, the enzyme activity changes of SOD, POD, and CAT were consistent with the changes of H_2_O_2_ content, and the DADS-treated leaves showed stronger ROS scavenging capabilities than the CK. This might suggest that the different accumulation of H_2_O_2_ may be a reason for the defense resistance difference between DADS treated and CK cucumber seedlings.

Previous studies reported that lignin protects cell wall polysaccharides from microbial degradation, and its biosynthesis was induced upon pathogen infection [20]. Therefore, we also check the lignin accumulations in the leaves of DADS-treated and CK cucumber seedlings after the infection of *P. cubensis*. The accumulations of lignin on the cell walls of cucumber leaves were truly induced when inoculated with downy mildew pathogens (Figure 2H), in which the leaf cells showed extension of lignification stained with toluidine blue (Figure 2H). During the infection process, the cell walls were gradually thickened, and their intensities were simultaneously deepened, especially at 4 hpi, and remained nearly unchanging since the 48 hpi (Figure 2H). However, the stained cell walls were not shown in CK leaves at 4 hpi (Figure 2H), and until the 96 hpi, the cell walls of CK leaves had similar performance with that of DADS-treated leaves at 48 hpi (Figure 2H). Therefore, DADS-treated leaves could accumulate lignin quicker than CK after the pathogen infection. Taken together, the activations of ROS and lignin accumulation pathways might take important roles in the DADS-induced cucumber resistance to *P. cubensis*.

### 2.3. DADS Induced Phytohormone Changes under the Infection of Downy Mildew

To determine whether phytohormone pathways were involved in the infection of *P. cubensis*, the contents of IAA, ABA, SA, and JA in cucumber leaves were measured by HPLC-ESI-MS/MS. The results showed that IAA levels in DADS-treated leaves were significantly increased and higher than those in CK leaves at 12, 48, and 72 hpi, while lower IAA content in DADS-treated leaves was detected at 0 hpi (Figure 3A). The contents of SA in DADS-treated leaves were significantly higher than those in CK leaves at 48, 72, and 96 hpi, and were significantly lower at both 12 and 168 hpi (Figure 3B). The SA content had a peak at 4 hpi for both the DADS-treated and CK leaves and then decreased and kept relatively steady until 96 hpi. Compared with the CK, the ABA content in DADS-treated leaves showed a significant fluctuation of higher to lower and then to higher levels (Figure 3C). The content of JA in DADS-treated leaves was significantly higher than that in CK leaves at 0 hpi, while, inversely, it showed significantly lower levels at both 24 and 48 hpi (Figure 3D). Similarly, when compared with the CK, the JA content in DADS-treated leaves also showed higher to lower and then higher levels from 0 to 168 hpi (Figure 3D).

### 2.4. DADS-Induced Significant DEGs in Cucumber to Improve its Downy Mildew Resistance

To obtain the potential related genes and pathways associated with the DADS induced resistance to *P. cubensis*, transcriptome profiling was performed with the *P. cubensis*-infected leaves that sampled at 0, 4, 24, and 48 hpi for both DADS-treated and CK cucumber seedlings. A total of approximately 97.7 GB clean reads were generated for 16 biological samples. The average GC content and Q30 values of the raw reads were 45.40% and 94.15%, respectively, indicating the high qualities of these reads, in which about 95% high-quality reads were successfully mapped onto the cucumber genome of Gy14_V2.0. (Appendix A). Further correlation evaluation showed that the gene expression levels among samples were consistent with the overall quality of the RNA-Seq data, indicating the reliability of the samples and experimental design (Figure 4A). DEGs were checked between the DADS treated and CK groups, in which 886, 1254, 1141, and 1475 DEGs were found for samples at 0, 4, 24, and 48 hpi, respectively. To independently assess the quality of the RNA-seq data, qPCR was performed to analyze the expression patterns of 10 disease resistance-related genes selected from the obtained DEGs. The high correlations (*r*^2^ = 0.7375) between the qPCR and RNA-seq data further implied the reliability of this RNA-seq data (Figure 4B). For the DEGs, the group of 0 hpi, CK-0h-vs-DADS-0h, had 481 up-regulated and 405 down-regulated genes; there were 677 up-regulated genes and 577 down-regulated genes in CK-4h-vs-DADS-4h; 697 up-regulated genes and 444 down-regulated genes were shown in CK-24h-vs-DADS-24h; and there were 1219 up-regulated genes and 256 down-regulated genes in CK-48h-vs-DADS-48h (Figure 4C). This suggested that DADS induced more up-regulated genes in responding to the infection of *P. cubensis*.

To know the specifically expressed DEGs and their involved pathways before and after the inoculation of downy mildew pathogen, Venn analysis was first conducted between the DEGs at the 0 hpi and the combined DEGs at 4, 24, and 48 hpi. As shown in Figure 5A, 310 and 1859 up-regulated DEGs were specifically expressed in the DADS-treated samples before and after the inoculation. Additionally, 171 DEGs were continuously expressed in the DADS-treated leaves either inoculated or not with pathogens (Figure 5A). KEGG pathway enrichment analysis was, respectively, performed for these three sets of up-regulated DEGs, in which the ‘plant hormone transduction’ pathway was significantly enriched among the 310 uniquely up-regulated DEGs at 0 hpi; the ‘phenylpropanoid biosynthesis’, ‘phenylalanine metabolism’, ‘mitogen activated protein kinases (MAPK) signaling’, ‘glutathione metabolism’, and ‘plant–pathogen interaction’ pathways were significantly enriched for 1859 specifically expressed DEGs at 4, 24, and 48 hpi; the ‘phenylpropanoid biosynthesis’, ‘phenylalanine metabolism’, ‘MAPK signaling’, and ‘plant–pathogen interaction’ pathways were significantly enriched in the 171 continuously expressed DEGs (Figure 5A). Furthermore, Venn analysis and the KEGG pathway enrichment analysis were also conducted for the up-regulated DEGs at different time points (4, 24, and 48 hpi) after pathogen inoculation. As shown in Figure 5B–D, the above pathways were also significantly enriched at different time points. The pathways of ‘phenylpropanoid biosynthesis’, ‘phenylalanine metabolism’, ‘MAPK signaling’, and ‘plant–pathogen interaction’ were significantly enriched at 4, 24, and 48 hpi (Figure 5B,C), while the ‘glutathione metabolism’ pathway was especially enriched at 24 and 48 hpi (Figure 5C). Additionally, the ‘plant hormone signal transduction’ pathway was significantly enriched at 4 and 24 hpi but not at 48 hpi (Figure 5B,C). Furthermore, the 83 shared DEGs by the time points of 4, 24, and 48 hpi were enriched in the pathways of ‘phenylpropanoid biosynthesis’ and ‘glutathione metabolism’ (Figure 5D). Similarly, we found that the down-regulated DEGs in the DADS-treated leaves were mainly enriched for the pathways of ‘sulfur metabolism’, ‘plant hormone signal transduction’, and ‘phenylpropanoid biosynthesis’. Moreover, ‘sulfur metabolism’ was significantly down-regulated, expressed at 4 and 24 hpi (Appendix A). Overall, DADS treatment can induce the expression of DEGs related to disease-resistance pathways in cucumber, which further improves the resistance of cucumber to the infection of pathogens.

Considering that the above-related pathways play vital roles in *P. cubensis* infection, the DEGs in these pathways were focused on checking their expressions between the DADS-treated and the CK leaves under pathogen infection. As the heatmaps show in Figure 6 and Appendix A, the DEGs involved in plant–pathogen interaction, MAPK signaling, glutathione metabolism, and phenylpropanoid biosynthesis pathway were continuously upregulated from 0 hpi to 48 hpi. Based on their KEGG annotation, there were 55 plant–pathogen interaction DEGs (3 *WRKY*, 3 *RBOH*, 16 *MYB*, 6 *HtpG*, 6 *CPK*, 8 *CML*, 2 *PR1,* and others) (Figure 6A, Appendix A), 25 MAPK signaling DEGs (3 *WRKY*, 2 *RBOH*, 3 *PYL4*, 2 *ERF*, 6 chitinase genes, and others) (Figure 6B, Appendix A), 23 glutathione metabolism DEGs (15 *GST* and others) (Figure 6C, Appendix A), 52 phenylpropanoid biosynthesis DEGs (6 β-glucosidase genes, 22 *POD*, 17 phenylalanine metabolism DEGs, and others) (Figure 6D, Appendix A), and 17 phenylalanine metabolism DEGs (10 *PAL*, 4 *4CL*, and others) (Figure 6E, Appendix A), respectively. Additionally, there were 16 DEGs related to sulfur metabolism (Figure 6F, Appendix A), in which 5 DEGs (1 *DIN1*, 1 *SULTR*, 1 *PDI*, 1 *yqjG*, and others) and 7 DEGs (1 *Apr1* gene, 1 *THI1*, and others) were, respectively, up–regulated and downregulated at 4 hpi and 3 DEGs (1 *LSU*, 1 *TRX2,* and 1 *NIR1*) were upregulated at 0 hpi. Moreover, DEGs involved in the plant hormone signal transduction were also obtained, including 16 IAA associated DEGs (9 *IAAs*, 3 *GH3*s, 2 *ARFs*, 1 *SAUR*, 1 *TIR1*) (Figure 6G, Appendix A), 9 ABA associated DEGs (7 *PYLs*, *1 SnRK2,* and 1 *ABF*) (Figure 6G, Appendix A), 4 SA associated DEGs (1 transcription factor *TGA* genes, 1 *EDS1*, and 2 *PR1* genes) (Figure 6I, Appendix A), and 12 JA associated DEGs (2 *MYC2*, 1 *JAR1*, 3 *AOS*, and 5 *JAZ*s) (Figure 6J, Appendix A).

### 2.5. WGCNA Provides Insights into Cucumber Resistance to P. cubensis

To further explore the hub genes involved in cucumber resistance to *P. cubensis*, WGCNA was conducted for the DEGs between CK and DADS treatments. For the up-regulated DEGs, 15 modules of highly correlated genes were obtained based on their expression levels and their hierarchical clustering of the topological overlap matrix (Appendix A). For the highlighted genes in the ‘turquoise’, ‘cyan’, ‘tan’, ‘blue’, ‘red’, ‘pink’, and ‘brown’ modules, the RPKM values were higher in the DADS-treated leaves at each sampling time point (Figure 7). RPKM values of genes in control samples were defined as their basal expression levels. Enrichment analyses of KO terms based on the seven modules revealed many genes that might be related to the resistance of DADS treated leaves. As shown in Figure 7, the KO terms enriched in the ‘turquoise’ module were ‘glutathione metabolism’ and ‘plant–pathogen interaction’ pathways; the ‘MAPK signaling’, ‘phenylalanine metabolism’, and ‘phenylpropanoid biosynthesis’ pathways were enriched in the KO terms of ‘blue’ module; the KO terms of ‘brown’ module were enriched with ‘MAPK signaling’ and ‘plant–pathogen interaction’ pathways; the KO terms of ‘pink’ module were enriched with the ‘MAPK signaling’ and ‘phenylpropanoid biosynthesis’ pathways; and the KO term enriched in ‘red’ module was ‘plant hormone signal transduction’ pathway (Figure 7). The correlations between the modules and the traits showed that the most positive correlation between gene significance (GS) and module membership (MM) of POD activity, H_2_O_2_, IAA, and SA content were corresponding to the ‘red’, ‘green’, and ‘black’ ‘magenta’ modules, respectively (Appendix A). Similarly, for the down-regulated DEGs, nine modules of highly correlated genes based on their expression levels were identified, in which the KO terms of the ‘phenylpropanoid biosynthesis’ pathway was enriched in the ‘pink’ module, and ‘plant hormone signal transduction’ pathway was enriched in ‘blue’, ‘green’, and ‘pink’ modules (Appendix A).

To determine the genes playing central roles in the regulating network, nine genes with high connectivity within each module, known as hub genes, were identified (Table 1). These hub genes were strongly enriched in metabolic process and plant–pathogen interaction pathways. Hub genes *CsPOD* (*CsGy4G012800*), *CsPAT* (*CsGy4G000360*), *CsPAL* (*CsGy6G023900*), *CsGST (CsGy1G004050*), and *CsPYL4* (*CsGy5G008430*) were significantly related (*** *p* < 0.001) (Appendix A) and had the highest connectivity in the regulating network and all of them were in the ‘blue’ module (Table 1). Additionally, other hub genes in the plant–pathogen interaction pathway, including *CsWRKY33* (*CsGy1G004610*), *CsGH3.6* (*CsGy6G010290*), *CsPR1* (*CsGy7G006240*), and *CsEDS1* (*CsGy1G001860*) (Table 1), were also found. The transcript levels of *CsGH3.6* (*CsGy6G010290*), *CsGST* (*CsGy1G004050*), *CsPAL* (*CsGy6G023900*), *CsPAT* (*CsGy4G000360*), and *CsPOD* (*CsGy4G012800*) were positively associated with that of *CsWRKY33* (*CsGy1G004610*) (Appendix A).

## 3. Discussion

### 3.1. Possible PTI and ETI Defense Mechanisms

To understand the mechanism of DADS-induced cucumber resistance to downy mildew, morphological observations and microscopic examinations of pathogen-infected leaves were performed for both the CK and DADS treatments. The results showed that DADS promoted the resistance of cucumber to downy mildew (Figure 1). Previous studies indicated that garlic extracts have broad-spectrum fungicidal effects against a wide range of fungi, including *Candida*, *Torulopsis*, *Trichophyton*, *Cryptococcus*, *Aspergillus*, *Trichosporon*, and *Rhodotorula* species. Additionally, DADS and diallyl trisulfide (DATS) separated from garlic essential oil also showed antifungal activity against some fungi (*C. albicans*, *C. tropicalis*, and *Blastoschizomyces capitatus*) [12]. Therefore, DADS and DATS are probably the major compounds of garlic extracts showing fungi resistance. To explore the potential genes related to the DADS-induced cucumber resistance to downy mildew, RNA-Seq was employed at different time points after the *P. cubensis* inoculation. The results of transcriptome profiling revealed that plenty of DEGs were putatively related to the DADS-induced cucumber resistance to *P. cubensis*. Numerous DEGs were involved in various defense responses that might be mediated by PAL, ROS, MAPK, GSH, SA, and IAA signaling pathways (Figure 5). This suggested that multiple processes are involved in cucumber to defend against downy mildew infection, which is consistent with the fact that plants have complex defense mechanisms [21,22].

At the first stage of infection, fungal pathogens secrete PAMPs that are recognized by pattern recognition receptors on host cell surfaces, resulting in pattern-triggered immunity (PTI) that can halt further colonization [22]. The activation of PTI results in a series of cellular responses, including the generation of ROS, changes in cytosolic ion flux, cascade activation of calcium-dependent or MAPK, and enhancement of physical barriers [23]. In this study, the genes related to phenylpropanoid biosynthesis, MAPK signaling, and phenylpropanoid metabolism (Figure 5) were found to be up-regulated in the DADS-treated cucumbers in responding to the infection of *P. cubensis*. Furthermore, plants can encode R proteins to activate effector-triggered immunity (ETI), which produces HR at the site of infection, causing cell death to prevent the further spread of pathogens [24,25]. Here, our work indicated that DADS treatments could generate ROS and induce HR (Figure 2) and significantly increase the expressions of ROS metabolic pathway-related genes (Figure 6) after *P. cubensis* infection. These results suggest that the invasion of *P. cubensis* triggered both PTI and ETI in DADS-treated cucumbers. This is different from the previous study, which reported that DADS inhibited plant PTI but induced ETI [9].

### 3.2. DADS Induced H_2_O_2_ and Lignin Response to P. cubensis Infection

The PTI and ETI both can generate ROS. The production of cellular ROS, including H_2_O_2_ and superoxide anion, can be induced by pathogen invasion [18]. In plants, lignin plays important role in resistance to pathogen infection [15]. In the present study, lignin and H_2_O_2_ significantly accumulated at the early stages of *P. cubensis* infection in DADS treated cucumbers (Figure 2B,C). The H_2_O_2_ accumulation might be balanced by its increased synthesis or decreased degradation [15]. For H_2_O_2_, its biosynthesis is mainly controlled by SOD, while its degradation is primarily controlled by POD and CAT. In this study, the activity changes of these related enzymes were consistent with the changes in H_2_O_2_ content. Among them, the activity of POD was always higher in the DADS-treated cucumbers compared with that of the CK ones (Figure 2F). In tomato, the activities of POD, CAT, and SOD in leaf and root were also increased after DADS treatment for 24 and 48 h [9]. Through RNA-seq analysis, 12 *POD* genes were detected in the phenylpropanoid biosynthesis pathway, which were significantly up-regulated at 4 hpi (Figure 5D). Additionally, the *POD* gene (*CsGy4G012800*) was found as a hub gene by WGCNA (Table 1). Moreover, DADS treatment also increased the expressions of *POD* genes in tomato, which further increased tomato resistance, probably through inducing and enhancing the ROS scavenging [9]. *POD* genes are also key genes of regulated lignin synthesis through the ‘Phenylpropanoid biosynthesis’ pathway [20]. Significantly increased lignin content was also found in DADS treated cucumbers under the *P. cubensis* infection (Figure 2H). In tomato, H_2_O_2_ induced the expressions of *PAL* genes in the roots of tomato plants showing disease resistance [19]. Taken together, the expression of H_2_O_2_ and lignin-related genes were affected by *P. cubensis* infection, which resulted in the rapid enrichment of H_2_O_2_ and lignin to increase the resistance of DADS treated cucumber plants.

### 3.3. DADS Induced GST Response to P. cubensis Infection

A recognized function of GSTs is their participation in antioxidative reactions, together with the pivotal cellular antioxidant GSH to eliminate ROS and lipid hydroperoxides that accumulated in infected tissues [26]. Eighteen *GST*s in the glutathione metabolism pathway were significantly up-regulated in the DADS-treated cucumbers (Figure 5C). Similarly, the ‘glutathione metabolism pathway’ is also the most significant pathway that was early induced by DADS in tomato roots, in which DADS increased the GSH content and GST activity [9]. Additionally, DADS could increase GST activity, GSH levels, and the expression of *Mrp2*, which mediates the transport of GSH-conjugates in male Sprague Dawley rats [27]. The expression of *GST*s was functionally characterized in *A. thaliana* plants in response to treatments with herbicides, phytohormones, oxidative stress, and the inoculation with virulent and avirulent strains of the obligate biotrophic downy mildew [26]. Furthermore, a *GST* was reported for the powdery mildew resistance in tomato [28]. *GST*s genes are strongly inducible by H_2_O_2_, SA, and IAA [29,30,31], in which some certain *GST* transcripts can be considered useful markers for increased intracellular availability of H_2_O_2_ [29], and some *GST*s showing auxin-inducible can bind auxins as non-substrate ligands as well as participate in auxin transport [31].

### 3.4. DADS Induced Plant Hormone Response to P. cubensis Infection

Phytohormones play essential roles in the regulation of plant stress responses, which include drought stress [32,33], biotic stress [34], and so on. In general, synthesis and perception of different stress hormones, along with their relative contents and interactions, seem to be crucial for plant resistance against pathogens [35]. For example, SA-responding pathway is associated with plant resistance to biotrophic and hemibiotrophic pathogens, while the JA signaling pathway plays an important role in resisting the invasion of necrotrophic pathogens [36]. In this study, compared with the CK, DADS-treated cucumbers presented higher contents of SA and IAA but lower contents of ABA and JA (Figure 3A–D) under the infection of *P. cubensis*, a typical biotrophic pathogen. This might suggest that these hormones are involved in the basal resistance of cucumber against downy mildew. SA is known to be crucial in basal defenses and SAR [37]. A previous study suggested the function of SA in response to *P. cubensis* infection [38]. Here, SA content was significantly increased in DADS-treated leaves since the 24 hpi (Figure 6I). Based on the RNA-seq data, several SA-associated genes, including *CsPAL*, *CsEDS1*, *CsTGA*, and *CsPRs* (Figure 6J) were upregulated, and expressions of several key genes involved in the JA biosynthesis and signaling pathways, including *CsJAR1*, *CsJAZ*, and *CsMYC2*, were also increased (Figure 6). In tomato, the interaction between JA and SA promoting disease resistance has also been reported [39]. In basal and in ETI, the *EDS1* gene promotes the accumulation of SA and is currently considered as the upstream gene in SA signaling [40]. This might inform us why the expression of *EDS1* was significantly increased in the DADS treated cucumbers under downy mildew infection.

SA is mainly synthesized by two pathways, the PAL pathway and the isochorismate synthesis pathways [41]. In this study, *CsPAL* was upregulated under the *P. cubensis* infection, which might suggest that SA is mainly synthesized via the PAL pathway in DADS treated cucumbers. ROS signals are involved both the upstream and downstream of SA signaling in response to stresses [42]. SA may facilitate H_2_O_2_ accumulation during the oxidative burst induced by infections of avirulent pathogens [37]. Interestingly, the study indicates that SA not only plays a pro-oxidant role but also has an antioxidant role in concert with GSH in response to stresses [42]. Additionally, the transcriptional control of the class II TGAs mediated by SA and ROS signals also has an essential role in plant defense responses to stresses [42].

Regarding auxins, little is known about their roles in plant resistance to biotrophs. Auxin might interact with the JA to improve plant resistance to necrotrophic pathogens [43]. However, in this work, IAA levels of the DADS-treated cucumbers were constantly higher than those of the CK (Figure 3A). Transcriptome profiling data revealed that DADS treatment induced the up-regulation of 16 IAA related genes (nine *IAA*s, three *GH3*s, two *ARF*s, one *SAUR*, and one *TIR1*) (Figure 6G). Auxin signaling has been connected to innate immunity, PAMP-triggered susceptibility, and PTI [44]. However, previous studies also reported that several plant pathogens can directly synthesize auxin or induce plant auxin biosynthesis or modulate auxin signaling to stimulate susceptibility [35]. Additionally, SA has been reported to induce the transcriptions of genes coding for IAA-conjugating enzyme GH3.5 that converts free IAA into inactive auxin [45]. In general, higher SA content reduces active IAA and represses auxin signaling, leading to the enhanced defense, while SA-mediated defenses are also attenuated by auxin [43,44]. Considering the higher basal levels of IAA and SA in the leaves of DADS treated cucumbers, IAA and SA might synergistically function in the resistant processes of quick responses to pathogen infections.

### 3.5. DADS Induced WRKY33 Response to P. cubensis Infection

In this study, several *WRKY* transcription factors were up-regulated; in particular, *CsWRKY33* was higher in the DADS-treated leaves at each sampling time point (Appendix A). Pathogen-induced *WRKY33* is an important transcription factor that regulates the antagonistic relationship between defense pathways mediating responses to *P. syringae* and necrotrophic pathogens [46]. Genetic analyses demonstrated that induced resistance of GO5 (Arabidopsis plants overexpressing glycolate oxidase in chloroplasts) is dependent on *WRKY33*, but not on camalexin production [47]. For *WRKY33*, its role in plant resistance to biotrophs is poorly known. Moderate production of H_2_O_2_ in chloroplasts can initiate the signaling events leading to the induction of *WRKY33* and its downstream target genes. The transcription levels of *CsGH3.6*, *CsGST*, *CsPAL*, *CsPAT,* and *CsPOD* were positively correlated with the expressions of *CsWRKY33* (Appendix A).

### 3.6. Possible Resistance Pathways in DADS Induced Cucumber Resistance to Downy Mildew

Based on the results of this study, we proposed a hypothetical model for interpreting the DADS induced cucumber resistance to *P. cubensis* (Figure 8). This model includes regulating genes involved in the IAA signaling, SA signaling, GSH metabolism, and ROS. Firstly, DADS treatment mainly induced the up-regulates of DEGs related to plant–pathogen metabolism, MAPK signaling, phenylpropanoid biosynthesis, phenylalanine metabolism, and plant hormone signaling pathways. The increased expressions of these DEGs promoted cucumber resistance and its responding effectiveness to the infection of pathogens. Then, the pathways of plant–pathogen metabolism, MAPK signaling, glutathione metabolism, phenylpropanoid biosynthesis, phenylalanine metabolism, and plant hormone signaling were activated when the DADS–treated cucumbers were infected with *P. cubensis*. Specifically, DADS entered cucumber cells via sulfur transporter (SULTF) and induced the accumulation of H_2_O_2_. Meanwhile, DADS also induced the PTI and ETI in cucumber against the *P. cubensis* infection. The H_2_O_2_ initiated the signaling to the inducing the expressions of *CsWRKY33* and its downstream target genes. H_2_O_2_ also triggered the SA, IAA, GST, and their related genes to defense against the infection of *P. cubensis*. Additionally, *GST*s genes were strongly inducible by H_2_O_2_, SA, and IAA. Simultaneously, SA might be mainly synthesized via the PAL pathway in the DADS-treated cucumber. In addition, IAA and SA might synergistically function in the resistant processes of quick responses to pathogen infections. Taken together, DADS increased the contents of H_2_O_2_, GST, SA, IAA, and lignin and induced the expressions of their related genes to promote the resistance of cucumber to *P. cubensis* infection. This research might provide a foundation for further studies on the mechanism of cucumber resistance to *P. cubensis* by DADS induced.

## 4. Materials and Methods

### 4.1. Cucumber Plant and P. cubensis Preparation

Downy mildew-susceptible cucumber inbred line CCMC (changchunmici), a North China fresh market type cucumber, was sowed in pots (7 cm × 7 cm × 10 cm) and grown in a growth chamber with a 16 h day/8 h night temperature of 25/18°C and relative humidity of 80%. Cucumber seedlings at the two-true-leaf stage were used for the following experiments. The *P. cubensi*s isolate used in this study was collected from infected cucumber leaves grown in a plastic tunnel of Northwest A&F University, Yangling (34°160 N, 108°40 E), China. Diseased leaves were incubated overnight in a moist container and washed with sterile water to collect sporangia (19.4~31.0 × 13.5~20.6 μm in size). Single sporangia were isolated and placed onto the whole leaves in a moist container for infection. After five to seven days of inoculation, propagated sporangia from the single sporangia-infected leaf tissues were collected. Subsequent pathogen propagation and maintenance were carried out by spraying a sporangia suspension (10^4^ sporangia/mL) on the leaves of intact plants. Inoculated plants were kept in a plastic tray with lids covered to maintain 100% humidity and kept in dark for 24 h and then moved to the growth chamber with the same settings for growing healthy cucumber plants described above [48].

### 4.2. DADS Treatment and P. cubensis Inoculation

To prepare the DADS stock solution, laboratory-grade DADS (purity 80%) was ordered from Sigma–Aldrich Co. (St. Louis, MO, USA) and first dissolved in Tween–80 with a ratio of 1:2 (*w/w*); then, distilled water was added to obtain a 10 mmol/L stock solution, which was stored at 4^o^C for further use [49]. The DADS stock solution was diluted to 1 mmol/L in distilled water for use in the following treatments. DADS solutions (1 mmol/L, 5 mL) were sprayed twice on leaves of each cucumber seedling with a 5–day interval, while an equal volume of distilled water was sprayed as the control. Three replications were performed for each treatment with 12 cucumber plants. After ten days of the first DADS treatment, 1 mL 1 × 10^5^ sporangia mL^−1^ solution of *P. cubensis* was sprayed onto the back of the second true leaf of each seedling for inoculation. The inoculated seedlings were first moved to the growth chamber under the following conditions: darkness 24 h, nearly 100% relative humidity, and with temperatures of 20 °C. Then, the light was provided in the growth chamber with a photoperiod of 16 h day/8 h night, while the day/night temperatures and the relative humidity were kept with 25/18 °C and 100%, respectively. The morphological changes of pathogen-inoculated seedlings were recorded by photography, in which the disease index was calculated based on the phenotypes (e.g., disease spots, yellowing) of each seedling after 7 days of inoculation. Leaf samples were, respectively, collected at 0 (DADS treated and uninoculated by *P. cubensis*), 4, 12, 24, 48, 72, 96, and 168 hpi (hours of post inoculation) for both the treatment and control. For each leaf sample, half was stored in the stationary liquids for histological observation of pathogen infection process and H_2_O_2_ and lignin accumulations, while the remaining half leaf was immediately frozen in liquid nitrogen and then stored at −80 °C for further biochemical assays and RNA sequencing analysis.

### 4.3. Histological Observation

To more clearly visualize the downy mildew infection process in cucumber, histological observations of the pathogenic sites and microscopic visualization of H_2_O_2_ and lignin accumulations or distributions were conducted with the DADS-treated and untreated cucumber leaves. For the observation of the pathogen infection process, as described by Savory et al. [50], leaf samples (1 cm^2^) were first decolored in 95% ethanol and then stained by trypan blue solution with a ratio of 1:1:1 for glycerol, lactic acid, and water. Visualizations were performed using an Olympus BX63 (Olympus, Tokyo, Japan) light microscope. H_2_O_2_ accumulations were checked by the 3,3-Diamino-benzidine (DAB)-staining method [51]. Briefly, cucumber leaf sections (1 cm^2^) were immersed in a 1 mg/mL DAB solution (pH 3.8) for 8 h under light conditions, and then the leaf samples were washed out with distilled water and observed under a microscope (Olympus BX63). For lignin accumulation analysis, cucumber leaf sections with an area of 1 cm^2^ were stained with toluidine blue solution (0.05%) for 10 min and then rinsed with distilled water for observation using a microscope (Olympus BX63) under white-light vision [52].

### 4.4. Biochemical Assays

Biochemical assays were conducted from the aspects of the contents of superoxide anion, H_2_O_2_, and phytohormones and the activities of defense enzymes using the frozen stored leaf samples at −80 °C. Measurement of superoxide anion content followed Wang et al. [53]. The content of H_2_O_2_ was checked with the method described by Gong et al. [54] using potassium iodide. To assay the activity of defense enzymes, cucumber leaves (0.5 g) were first weighed and ground together with 6 mL of 200 mmol/L phosphate buffer (pH 7.8) containing 1% (*w*/*v*) soluble polyvinyl pyrrolidone (PVP) under ice bath condition. Then, the homogenates were centrifuged at 12,000 rpm and 4 °C for 20 min, and the supernatants were collected to measure the activities of superoxide dismutase (SOD), catalase (CAT), peroxidase (POD), and phenylalanine ammonia lyase (PAL) [3]. Auxin (IAA), abscisic acid (ABA), SA, and JA were extracted and purified following the procedures described by previous studies [55,56]. Measurement of these purified phytohormones was conducted with the HPLC-ESI-MS/MS (QTRAP 5500, AB Sciex, Boston, MA, USA), as described by Liu et al. [57].

### 4.5. RNA Extraction, cDNA Library Construction and Sequencing

When cucumber seedlings were inoculated with downy mildew pathogens, leaf samples of both DADS-treated and water-treated (CK) seedlings were collected at the following time points of 0, 4, 24, and 48 hpi with 2 biological replicates. In total, 16 leaf samples were obtained and used for the RNA extraction and sequencing. Total RNA was extracted using a Trizol reagent kit (Invitrogen, Carlsbad, CA, USA) following the manufacturer’s protocol. The qualities of extracted RNA were assessed on an Agilent 2100 Bioanalyzer (Agilent Technologies, Palo Alto, CA, USA) and checked using RNase-free agarose gel electrophoresis. Then, the mRNA, enriched by Oligo (dT) beads, was fragmented into short fragments using fragmentation buffer and reversed transcript into cDNA with random primers. The second-strand cDNA was synthesized by DNA polymerase I, RNase H, dNTP, and buffer. Subsequently, the cDNA fragments were purified with QiaQuick PCR extraction kit (Qiagen, Venlo, The Netherlands), end-repaired, poly (A) added, and ligated to Illumina sequencing adapters. The ligation products were size selected by agarose gel electrophoresis, PCR amplified, and sequenced using Illumina HiSeq2500 by the Gene Denovo Biotechnology Co. (Guangzhou, China). All clean reads generated in this study were deposited in the NCBI Sequence Read Archive database (http://www.ncbi.nlm.nih.gov/sra/, accessed on 3 November 2021) under the project accession number PRJNA776553.

### 4.6. Differentially Expressed Genes (DEGs) Analysis

The obtained clean reads were mapped to the cucumber genome of Gy14_V2.0 (ftp://cucurbitgenomics.org/pub/cucurbit/genome/cucumber/Gy14/V2//, accessed on 14 May 2019) by the alignment program of HISAT2 [58]. For each transcription region, an FPKM (fragment per kilobase of transcript per million mapped reads) value was calculated to quantify its expression abundance and variations, using StringTie software. DEGs analysis was performed by the DESeq2 software, in which the false discovery rate (FDR) < 0.05 and |fold change (FC)| ≥ 2 were used as the cut-off for considering the significant DEGs.

### 4.7. Expression Analysis of Disease Resistance-Related Genes

To preliminarily verify the accuracies of the DEGs obtained by RNA-sequencing, 10 resistance-related genes were selected from the DEGs list for qPCR analysis, including 6 ROS metabolism-related genes (*CsAPX*, *CsMDAR*, *CsGDX*, *CsSOD*, *CsPOD*, and *CsCAT*), 2 *DIR1* genes, and 2 major resistant genes of downy mildew (*CsSGR* and *CsSGRL*). The cDNA templates used for qPCR were the same as those used in the RNA sequencing. The qPCR analysis was conducted using 2 × SYBR Green III qPCR Mix (JIEYI Biotechnology Co., Ltd) on a QuantStudio^®^5 Real-Time PCR System (Life Technologies, Carlsbad, CA, USA). The cucumber Ubiquitin extension protein (UBI-ep) gene was used as the internal reference gene. Detailed sequence information of the primers used for qPCR were listed in Appendix A. Gene expression levels were calculated based on the 2^−^^ΔΔCt^ method [59]. Then, log2(FC) of the 10 genes were calculated and compared with RNA-sequencing.

### 4.8. Analysis of KEGG Enrichment of DEGs

Significant enrichment pathways were analyzed in the Kyoto Encyclopedia of Genes and Genomes (KEGG) database (http://www.genom e.jp/kegg/, accessed on 15 May 2019), in which a false discovery rate (FDR)-adjusted *p*-value ≤ 0.05 was used to indicate that the DEGs were significantly enriched in a pathway.

### 4.9. Weighted Gene Co-Expression Network Analysis (WGCNA)

Gene co-expression networks were constructed using the WGCNA analysis in the Omicsmart analysis platform of Gene Denovo Biotechnology (https://www.omicshare.com/WGCNA/home.html, accessed on 13 March 2021), in which the up-regulated and down-regulated DEGs were analyzed, respectively. Module–trait associations were estimated using the correlation between the module and the POD, H_2_O_2_, IAA, and SA of DADS/CK treatments.

### 4.10. Statistical Analysis

Randomized block design with three replications was for the involved experiments. Statistical analyses were performed using SPSS 13.0 (IBM, Armonk, New York, NY, USA). The significant differences were determined by one-way ANOVA, *t*-test, with * *p* < 0.05 and ** *p* < 0.01.

## Figures and Tables

**Figure 1 ijms-22-12328-f001:**
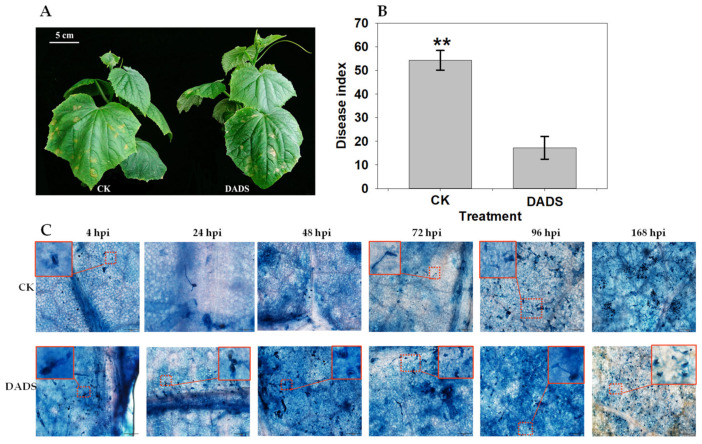
Comparison of leaf morphology (**A**), disease index (**B**), and process of *P. cubensis* infection (**C**) between DADS treated and CK cucumber seedlings. The inserted box in the top corner is a magnified image of the selected area. Data are means ± standard errors (*n* = 3, three biological replicates). Asterisks indicate significant differences between DADS treatment and CK (** *p* < 0.01) based on a one-way ANOVA followed by a *t*-test. Scale bars: 5 cm (**A**), 100 μm, 20 μm (**C**).

**Figure 2 ijms-22-12328-f002:**
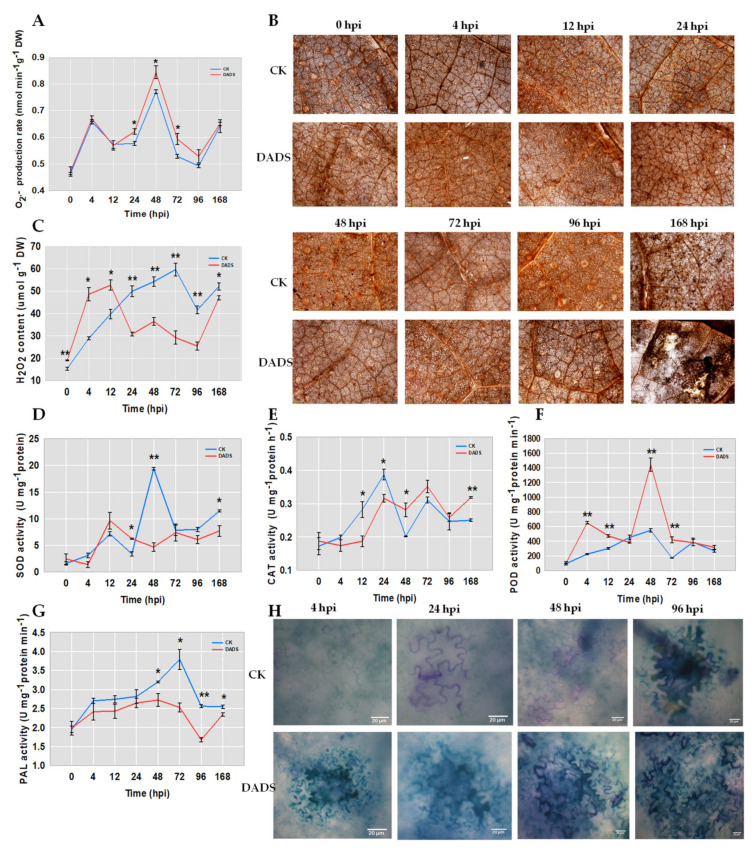
Comparison of ROS (**A**–**C**), defense enzyme (**D**–**G**), and lignin (**H**) in DADS-treated and CK cucumber leaves with *P. cubensis* infection. (**B**) Histological observation of H_2_O_2_ by DAB dyed. (**H**) Histological observation of lignin by TB dyed. Data are means ± standard errors (n = 3, three biological replicates). Asterisks indicate significant differences between the DADS treatment and CK (* *p* < 0.05; ** *p* < 0.01) based on a one-way ANOVA followed by a *t*−test. Scale bars: 200 μm (**B**), 100 μm (**H**).

**Figure 3 ijms-22-12328-f003:**
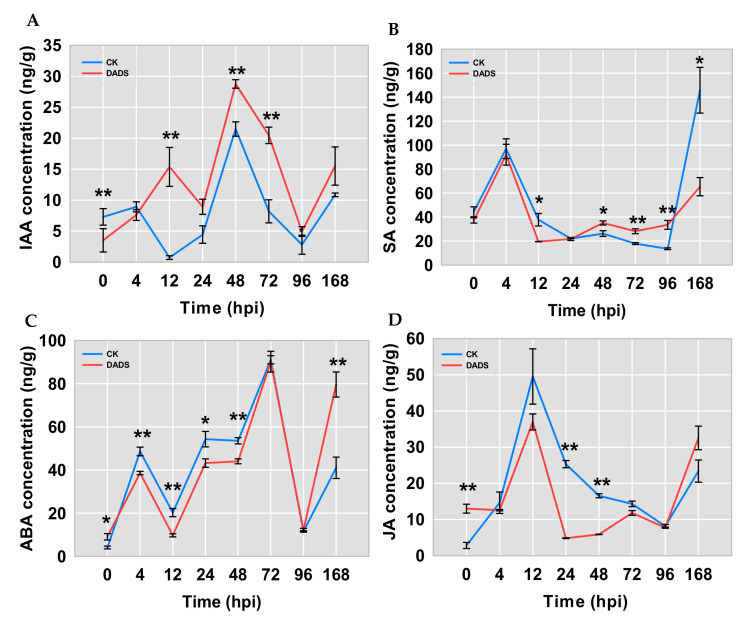
Comparisons of the IAA (**A**), ABA (**B**), SA (**C**), and JA (**D**) content in DADS-treated and CK leaves with *P. cubensis* infection. Data are means ± standard errors (n = 3, three biological replicates). Asterisks indicate significant differences between the DADS treatment and CK (* *p* < 0.05; ** *p* < 0.01) based on a one-way ANOVA followed by a *t*−test.

**Figure 4 ijms-22-12328-f004:**
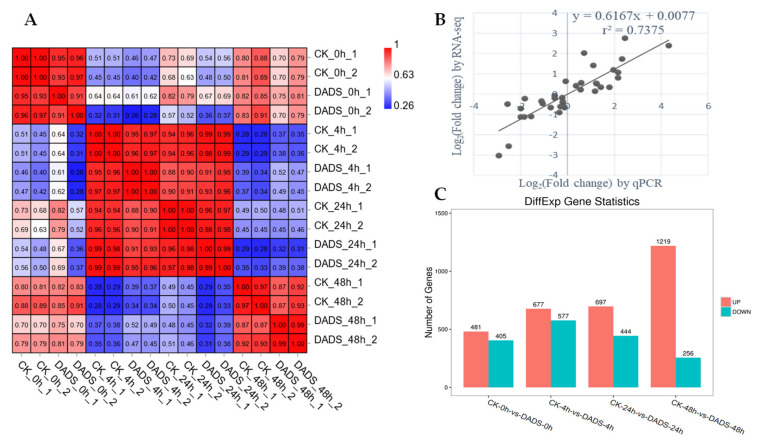
RNA−seq data and DEGs in DADS-treated and CK leaves with *P*. *cubensis* infection. (**A**) Pearson correlation of 16 samples based on the correlation coefficient (*r*^2^) between each sample. The color panel represents the *r*^2^ values. (**B**) Correlation of expression changes observed by RNA−seq (*y*−axis) and qPCR (*x*−axis). (**C**) Number of up− and down−regulated DEGs for each sampled time point. Up and downregulated DEGs are displayed in red and green bars, respectively.

**Figure 5 ijms-22-12328-f005:**
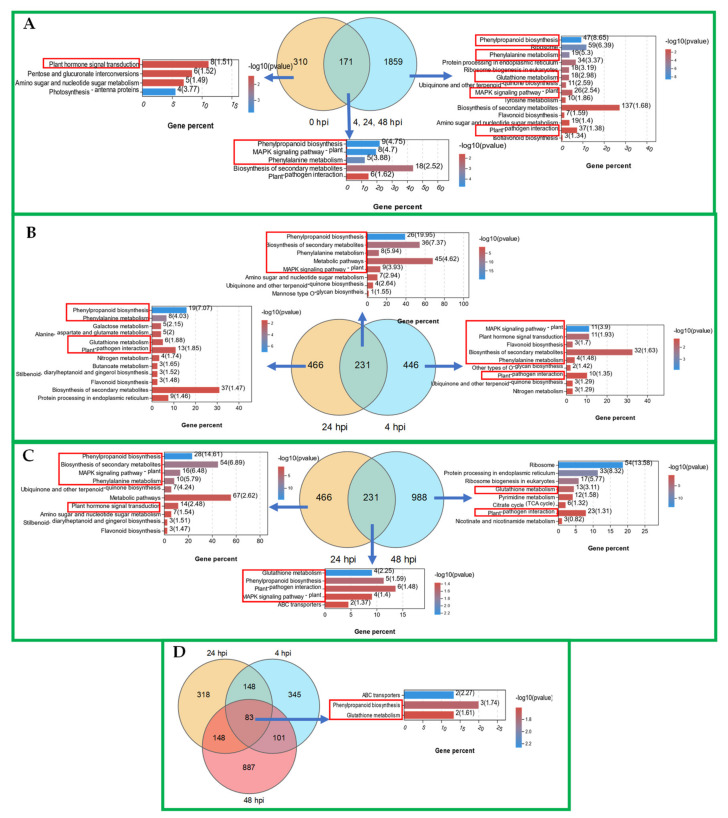
Venn analysis and KEGG enrichment analysis of up−regulated DEGs.

**Figure 6 ijms-22-12328-f006:**
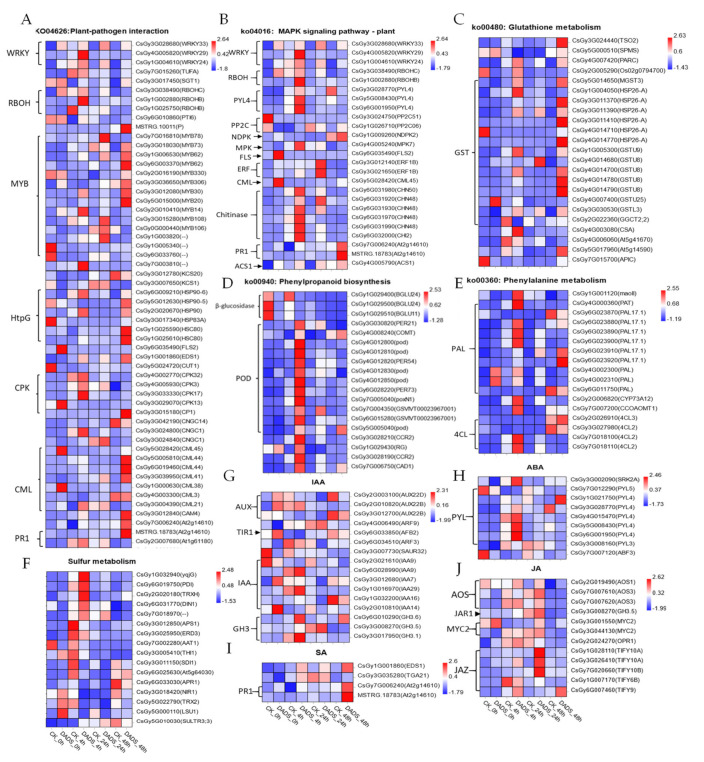
DEGs heatmap of DADS treated and CK leaves with *P. cubensis* infection. The bar represents the scale of the expression levels for each gene (FPKM) in the different treatments, as indicated by red/blue rectangles. Genes in red show upregulation, and those in blue show downregulation. (**A**) Plant–pathogen metabolism, (**B**) MAPK signaling pathway, (**C**) glutathione metabolism, (**D**) phenylpropanoid biosynthesis, (**E**) phenylalanine metabolism, (**F**) sulfur metabolism, (**G**) IAA genes, (**H**) ABA genes, (**I**) SA genes, and (**J**) JA genes.

**Figure 7 ijms-22-12328-f007:**
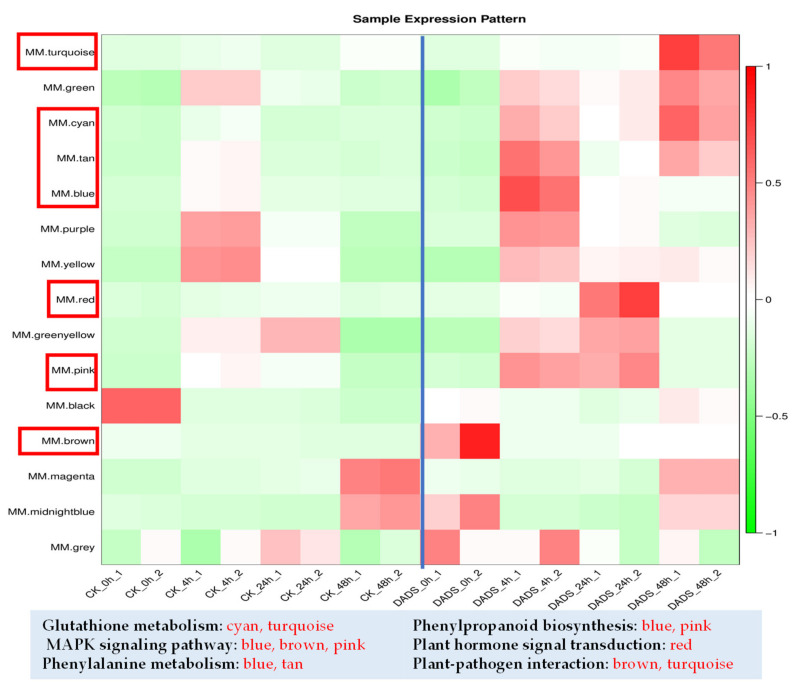
Expression modules constructed by WGCNA and associated KEGG of up–regulated DEGs in the ‘turquoise’, ‘cyan’, ‘tan’, ‘blue’, ‘red’, ‘brown’, and ‘pink’ modules.

**Figure 8 ijms-22-12328-f008:**
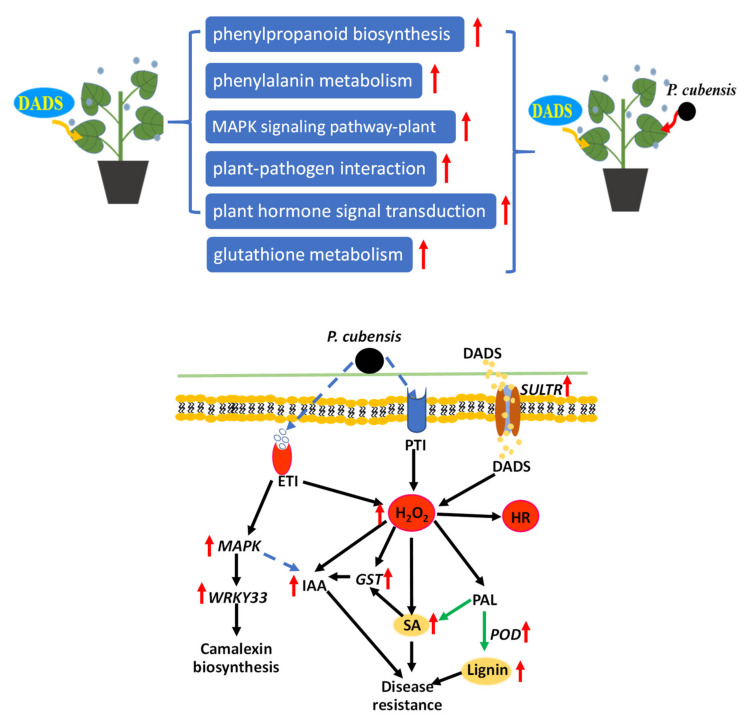
A potential model underlying the enhanced cucumber resistance to *P. cubensis* induced by DADS. Under the infection of *P. cubensis*, DADS-treated plants exhibit enhanced disease resistance in comparison with CK plants due to the activation of multifaceted defense machinery in leaves. Red arrows represent up-regulation, green arrows represent synthesis, black arrows represent facilitation, and dashed arrows represent unknown.

**Table 1 ijms-22-12328-t001:** Disease resistance-related hub genes revealed by WGCNA.

Gene	Description	Module Colors	All.k within
*CsGy4G012800*	Peroxidase	blue	179.41
*CsGy4G000360*	Aminotransferase	blue	178.46
*CsGy6G023900*	Phenylalanine ammonia-lyase	blue	146.39
*CsGy1G004610*	probable WRKY transcription factor 33	pink	12.23
*CsGy1G004050*	probable glutathione S-transferase	blue	104.07
*CsGy6G010290*	indole-3-acetic acid-amido synthetase GH3.6	tan	39.20
*CsGy5G008430*	abscisic acid receptor PYL4	blue	111.72
*CsGy7G006240*	PR1, Cysteine-rich venom protein	turquoise	27.60
*CsGy1G001860*	protein EDS1L-like	cyan	17.28

## Data Availability

The RNA-seq detail data in NCBI Sequence Read Archive database https://www.ncbi.nlm.nih.gov/bioproject/?term=PRJNA776553 (accessed on 22 October 2021).

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
