# Peer review of "Garlic Volatile Diallyl Disulfide Induced Cucumber Resistance to Downy Mildew"

_ijms, 2021, doi:10.3390/ijms222212328_

Round 1

Reviewer 1 Report

Manuscript by Yang et al. on “Garlic Volatile Diallyl Disulfide Induced Cucumber Resistance to Downy Mildew” used application of DADS preventatively to study the impact on enhancing cucumber resistance to downy mildew. Although this study has identified novel findings helpful for enhancing host resistance with preventative control measures, some aspects of the manuscript need improvement and the following comments might help in making this manuscript comprehensive.

Major comments:

  1. The figures included in the manuscript are of low quality and not fully legible. High quality figures are extremely important for submission
  2. How was the pathogen isolated and pure culture obtained? No information on the pathogen isolate has been provided.
  3. Statistical analysis needs more information. How was the pairwise comparison done for screening and qRT-PCR analysis? What was the experimental design used in the experiment?
  4. RNAseq sequencing design and analysis are very succinct and needs more elaboration. Sequencing data is completely missing. Is the data used in the manuscript publicly available at the time of manuscript submission? Deposit the data in public archive and mention the location of the data in the manuscript.
  5. Application of DADS precede pathogen inoculation. This makes DADS as preventative control. Why was this workflow considered rather than post-application of DADS after pathogen inoculation? DADS can also enhance the defense responses like hydrogen peroxide production. How was the defense responses/ROS production by DADS application and pathogen inoculation discerned in the study? The preventative control measure proposed in the study should also be reflected in the manuscript.
  6. There are several language errors in the manuscript which need critical attention.

Minor comments:

Line 519: Also mention about the working solution preparation of DADS

Line 522: What was the total volume of inoculation: “After ten days of the first DADS treatment, 1 x 105 sporangia mL-1 solution of P. cubensis were sprayed onto the back of the second true leaf of each seedling for inoculation.”

Line 524: moved

Line 525: How long were the plants kept in the dark after inoculation?

Line 530: chlorosis

Line 590: Confirm “absolute fold change” or just “fold change”

Line 593: How was the selection of 10 putative R-gene candidates done for qRT-PCR analysis?

Reviewer 2 Report

The authors report an interesting study on the action of diallyl disulfide (DADS) in Cucumber cultures. In particular, the authors identify molecular mechanisms underlying resistance to P. cubensis infection.

In particular, after a treatment with DADS (foliar nebulization), the authors monitor the response to P. cubensis infection and report the plant's ability to resist infection in histological and structural terms and by evaluating the synthesis of hormones and other defensive responses. Finally, the authors identified the potential genes responsible for the P. cubensis attack response

In general, the manuscript is very interesting, well written and clear. The authors define their hypotheses and the objectives of the manuscript well and the results obtained are remarkable and well discussed. Finally, I consider the statistical approach to be correct.

However, I have some suggestions for the authors:

  • No abbreviations should be used in the abstract. POD, IAA and SA should be made explicit.
  • The introduction is well structured and the authors clearly define the problem. I suggest to insert a few sentences on the effects of some hormones also in other conditions of environmental stress (see: Khaleghnezhad V et al. Concentrations-dependent effect of exogenous abscisic acid on photosynthesis, growth and phenolic content of Dracocephalum moldavica L. under drought stress. Planta . 2021 May 25; 253 (6): 127. Doi: 10.1007 / s00425-021-03648-7; and Naservafaei S, et al. Biological Response of Lallemantia iberica to Brassinolide Treatment under Different Watering Conditions. Plants (Basel). 2021 Mar 5;10(3):496. doi: 10.3390/plants10030496)
  • Authors need to improve the quality of the figures, it was really difficult to review the manuscript without fully understanding the gene names or histological details.

After these small changes I believe the manuscript is a great work to be published on IJMS

Author Response

Dear Editors and Reviewers,

Thank you for your great comments about our manuscript entitled “Garlic Volatile Diallyl Disulfide Induced Cucumber Resistance to Downy Mildew” (ID: ijms-1452717). These constructive comments are really valuable and helpful for improving our manuscript. We have carefully considered all the comments and accordingly revised our manuscript. Please see all the modified details in the revised manuscript that tracked all the changes. We hope that our revised manuscript will satisfy the quality of IJMS. Following listed the one-by-one responses to reviewers’ comments:

The authors report an interesting study on the action of diallyl disulfide (DADS) in Cucumber cultures. In particular, the authors identify molecular mechanisms underlying resistance to P. cubensis infection.

In particular, after a treatment with DADS (foliar nebulization), the authors monitor the response to P. cubensis infection and report the plant's ability to resist infection in histological and structural terms and by evaluating the synthesis of hormones and other defensive responses. Finally, the authors identified the potential genes responsible for the P. cubensis attack response

In general, the manuscript is very interesting, well written and clear. The authors define their hypotheses and the objectives of the manuscript well and the results obtained are remarkable and well discussed. Finally, I consider the statistical approach to be correct.

However, I have some suggestions for the authors:

  1. No abbreviations should be used in the abstract. POD, IAA and SA should be made explicit.

Response: We have corrected this in our revised manuscript. “Furthermore, both lignin and H2O2 were significantly increased by DADS treatment to responding P. cubensis infection. Simultaneously, the enzyme activities of peroxidase (POD) in DADS treated seedlings were significantly promoted. Meanwhile, both the auxin (IAA) and salicylic acid (SA) contents were increased and their related differentially expressed genes (DEGs) were up-regulated when treated with DADS.”

  1. I suggest to insert a few sentences on the effects of some hormones also in other conditions of environmental stress (see: Khaleghnezhad V et al. Concentrations-dependent effect of exogenous abscisic acid on photosynthesis, growth and phenolic content of Dracocephalum moldavica L. under drought stress. Planta . 2021 May 25; 253 (6): 127. Doi: 10.1007 / s00425-021-03648-7; and Naservafaei S, et al. Biological Response of Lallemantia iberica to Brassinolide Treatment under Different Watering Conditions. Plants (Basel). 2021 Mar 5;10(3):496. doi: 10.3390/plants10030496)

Response: These articles were very valuable and useful for us. We have carefully read and cited in the revised manuscript. “Phytohormones play essential roles in the regulation of plant stress responses, which includes drought stress [32, 33], biotic stress [34], and so on.”

  1. Authors need to improve the quality of the figures, it was really difficult to review the manuscript without fully understanding the gene names or histological details.

Response: We have adjusted the resolution of all the figures and uploaded the original high-quality images.

Reviewer 3 Report

This article is interesting, useful and well prepared. This is a carefully done study and the findings are of considerable interest about "Garlic Volatile Diallyl Disulfide Induced Cucumber Resistance to Downy Mildew". 

Introduction it is in line with the Instructions for the Authors. The methodology is also corresponding to the experimental part points of interest. Results of this study might give new hits on exploring the induced resistance mechanism of cucumber to downy mildew and provide useful information for the subsequent mining of resistance genes in cucumber.

It is an interesting research work and it is recommended to publish it as a first from a series with incoming findings.

Yours sincerely,

Author Response

Dear Editors and Reviewers,

Thank you for your great comments about our manuscript entitled “Garlic Volatile Diallyl Disulfide Induced Cucumber Resistance to Downy Mildew” (ID: ijms-1452717). These constructive comments are really valuable and helpful for improving our manuscript. We have carefully considered all the comments and accordingly revised our manuscript. Please see all the modified details in the revised manuscript that tracked all the changes. We hope that our revised manuscript will satisfy the quality of IJMS. 

This article is interesting, useful and well prepared. This is a carefully done study and the findings are of considerable interest about "Garlic Volatile Diallyl Disulfide Induced Cucumber Resistance to Downy Mildew".

Introduction it is in line with the Instructions for the Authors. The methodology is also corresponding to the experimental part points of interest. Results of this study might give new hits on exploring the induced resistance mechanism of cucumber to downy mildew and provide useful information for the subsequent mining of resistance genes in cucumber.

It is an interesting research work and it is recommended to publish it as a first from a series with incoming findings.

Response: Thank you for your positive feedback and recognition of our work.

Round 2

Reviewer 1 Report

I would like to thank Yang et al. for making the recommended changes in the manuscript. 

  1. I could not access the bioproject in NCBI for the RNAseq data which may be still in process of upload or approval by NCBI. However, it should be publicly available before the manuscript can be accepted.
  2. In figure 8, please change H2Oand make sure genes notations are italicized
  3. I would still recommend authors to provide isolate number for the pathogen and specific details on pathogen used in the study so other can access it whenever required. 
